# Searching for the Center: A New Civic Role for the Central Business District in China

**Yiyong Chen [1,2,\*]**, **John Zacharias [3,\*] and Mali Zeng [1,2]**

1   Shenzhen University, School of Architecture and Urban Planning, Shenzhen Key Laboratory of Built Environment Optimization, Shenzhen 518060, China
2   Key Laboratory of Urban Land Resources Monitoring and Simulation, Ministry of Land and Resources, P.R.C., Shenzhen 518060, China
3   Peking University, Laboratory for Urban Process Modelling and Applications, Beijing 100871, China
*   Correspondence: chenyiy@szu.edu.cn (Y.C.); johnzacharias@pku.edu.cn (J.Z.); Tel.: +86-755-26732858

**Abstract:** The central business district (CBD) has become the economic powerhouse of contemporary cities. China's economic transition from world factory to a knowledge-based economy underpinned the development of hundreds of CBDs over the course of less than two decades. The plans promoted land use diversity and the incorporation of service facilities in the support of business function, but a rather different service environment emerged. Taking the Futian CBD of Shenzhen as the prototypical case, we examined the distribution, vitality, uses, and users of these facilities, which are largely built up by the private sector and without governmental support. A questionnaire sent to users and data derived from social media reveal that the vast majority of visitors of these service facilities do not work in the CBD and travel via the reformed mass transport system to this location. The high-quality public spaces and street environment, as well as the numerous service facilities, many of which are at a low economic order, attract people from all over the vast city, which homes over ten million, highlighting a new role for the CBD as a civic center. In contrast with the globalized business sought after by government and business leaders of the CBD, a new populist nexus is emerging and without significant support.

**Keywords:** non-business activity; civic center; service facilities; Futian CBD; Shenzhen; vibrancy

## 1. Introduction

Cities have always had centers, although the purpose, design, and activities taking place there have changed over the course of urban history. The representational function of a central place as a symbol of the city was typically associated with decision-making at the level of the city, as agora in Hellenic culture, forum in Roman civilization, square and palace in Renaissance Europe, and most recently the Central Business District (CBD) of North American cities [1–3]. The daily activities of these significant places in the city might vary over time and with the role of the city in the region, but power and decision-making were always invested there. Contemporary CBDs in North America mostly sprang up at the geographical and historical heart of the city, with the surviving elements of the civic, religious, and cultural institutions. The CBDs in the Western Hemisphere assumed that these historical civic functions were growing the corporate command and decision-making function. China, with more planned CBDs currently under development than in all other countries in the world combined, has followed a somewhat different development trajectory. The question arises whether the purpose and functions of these new and highly symbolic centers for the city are fulfilled in the patterns of daily life. In particular, does the civic function of the center as a place for public gathering, cultural representation, communication, and leisure arise in the CBD or elsewhere in the Chinese city

when it was never intended to do so? Exploring these issues is not only of typical significance to the development of China's future CBD but can also can provide a reference for the renewal of western CBDs and for the CBD construction in former socialist cities.

Casual observation suggested that the transformation into a civic role was occurring naturally but required further investigation of the observed users to evaluate whether this new people presence constituted a transformation into a civic role. We conducted surveys of visitors to the CBD facilities, to understand who they are, where they reside, and what they are doing in the CBD, using the city of Shenzhen as a case study, which was hardly unique in this respect. Are the non-business facilities of the CBD serving the core business elite, or are they acquiring a widely shared public role in the city, more akin to the traditional roles of the city center, as mentioned above? If the latter, then how can we conceive the further transformation of the CBD in the contemporary city in China?

## 2. Literature

### 2.1. CBD Development in America

The American CBD is the physical prototype of the CBD in China, but it has quite a different trajectory and outcome. In the United States, the CBD prospered in tandem with suburbanization, but began a slow decline, starting in the 1960s, towards becoming the sole center for tertiary employment, particularly following the massive development of the interstate highway system. The command hub function of the CBD was largely retained, although retailing, services, and public functions mostly disappeared from the core [4]. Although CBD is still the core agglomeration area of urban economic activities, the decay trend seems unstoppable [5]. For example, in Los Angeles, the post-modern city is said to be represented by a centerless urban form [6]. American CBDs are generally uninhabited, with highest value residential areas in certain inner suburbs or exurbia. The focus on business activities, strongly supported by the urban plans of the 1960s and 1970s, resulted in the loss of services, shopping and public activity in city centers, an outcome that has been the focus of many recent efforts at regeneration. The sustained development of suburban employment centers, often accounting for much more employment than the traditional CBD, exacerbated these issues [7].

Because of the high land value, people seldom live in CBDs; thus, during the nighttime and rest days, CBDs become ghost towns, which also brings a series of security issues. Queries were aimed at the Manhattan-style CBD which was characterized by skyscraper office buildings and a lack of residential buildings, resulting in deserted streets after office hours. American cities are undergoing spatial restructure, resulting in the great majority of workplaces no longer being in CBDs and employment sub-centers [7]. Since the 1990s, the CBD revitalization movement, which brought CBD to a new stage and began in North America, spread to European cities. CBDs have evolved in competition with subcenter or edge cities to maintain its traditional economic role and importance [8]. With the coming of the post-industrial era, informative technology and service are becoming more and more important in the urban economy, especially in CBD [9]. The CBD revitalization movement, incorporating economic development, the public realm, and social, cultural, and transport strategies and initiatives, aims to make the CBD a better place to work, live, shop, play, visit, and stay [10,11]. A lot of new commercial buildings are built in CBDs, accompanied by shopping malls, retail centers, upscale apartments, and cultural facilities. Because of an infusion of resident-friendly concepts, such as gentrification and commercial, recreational, and cultural introduction, bringing vivid nightlife back to the downtown core, the CBD gradually transforms from a single office agglomerated area to an integrated urban center with rich facilities and various activities [12].

### 2.2. CBD in China

The typical urban plan in China in the 1980s saw the city divided into districts with factories, schools, community facilities, and shopping facilities, including shopping centers [13]. There was no CBD but rather workplaces distributed across the city in local centers and industrial districts in an effort

to reduce commuting travel. In theory, life could be led, for the most part, locally. The contemporary planned city in China is, in this sense, in continuity with the socialist city, conceived as an agglomeration of work units, each with the full range of community facilities and workplaces in a single physical space—a city without a center.

The lead-up to China's accession to the World Trade Organization in 2001 fundamentally changed the way people thought about the urban plan. Cities reformed their existing urban plans to make the CBD. The CBD was typically planned at the city edge as a place exclusively devoted to business activities. Rooted in a different social background and development mode, China's CBDs are substantially different in function and spatial structure when compared with western cases [14]. Planned development took place at the city's peripheries, supported initially by a reformed road network and then by mass public transport [15]. As a consequence of this, its development has been unrestrained by the existing city and the necessary redevelopment from the beginning [16]. Some CBDs in China were originally planned to include non-business functions and activities to avoid the problem of a nine-to-five environment. Exploratory attempts were made to merge various service facilities with official buildings, which have played an important role in the revitalization of the downtown area [17]. The new CBDs of Beijing, Shanghai, Guangzhou, and Shenzhen were centrally located in the transport system of their respective cities. High culture facilities joined the office functions in the core area. The CBD as a place for leisure, recreation, and popular culture was absent from the plan, notably with large, formal open spaces designed to serve as a symbolic setting for administrative and office buildings, not solely as places for public gathering. The CBD of China is undergoing a transformation into a new civic role.

As an important embodiment of urban modernization, China's Central Business District (CBD) is a well-planned entity that is dedicated exclusively to the workday needs of business elites [18]. The CBD aims to create a highly accessible space that leverages regional command, finance, and information economy functions. In China, the construction of modernity is considered necessary to divide its spatial scope, apply uniformly to certain urban design rules, and determine the scope and combination of mixed activities [19], including commercial and cultural roles. For these purposes, compared to American CBDs, which were typically established in central urban areas, China's CBDs are mostly developed in traditional urban fringe areas, similar to European cases, such as La Defense in Paris and Moscow City in Moscow [20]. The development of the CBD kept a distance from the historical cities and was largely unaffected by previous urban land uses.

Modernity is elusive and eager, and perhaps more desirable, because it is difficult to grasp and define [21]. Seeking modernity in China is a sub-text of almost all major government initiatives that are particularly prominent in urban development because it can be obvious and fast. Contemporary modernity responds to globalization by focusing on economic structures [22] rather than the cultural or political issues in some socialist cities [23,24]. Although economic modernization can be said to focus on commercial areas, the CBD project goes far beyond these needs. In the process of the globalization of large cities in China, no large-scale project can be compared with the new CBD construction.

China's dream of a modern city core dates back to the 1950s, although the term "CBD" did not appear in official planning discourse until the 1990s [18]. All major cities in China have hosted design competitions for these CBD developments, attracting extensive international participation [25]. The graphic form of the CBD plan occupies a vast space at the edge of the city and becomes a new hub for a larger urban-scale sports system. Although the form and design expression in the new CBD is late modernism, the idea behind urban planning is entirely modernist, positioning the synchronization of social and spatial orders [26]. A lot of non-business function and activities were incorporated in the master plan of CBDs from their original development. These exploratory attempts, which were made to merge various service facilities with official buildings, are expected to play a similar role to that of downtown area revitalization, as shown in western cases [17]. After a fast development period of about 20 years, it is time to consider the results of these herculean efforts under the context of global CBD development.

### 3. The Case of Shenzhen

All of China's first-tier cities began to launch the construction of CBDs in the 1990s. The CBDs were planned to integrate non-business activities into the CBD, including commercial, cultural, and residential uses, among others. The Futian CBD of Shenzhen was chosen as our study case as a typical case among the first-tier cities in China. The Futian CBD was the first attempt at merging the administrative center and high cultural institutions into the CBD. Being a lead case, its attempts threw light on many CBDs that were developed latter, such as Guangzhou, Hangzhou, Suzhou, and Zhengzhou, etc. Many of its practices and experiences are also deeply affected the subsequent CBD construction in China. Recently, as Shenzhen has been established as a pioneering demonstration zone of socialism with Chinese characteristics, its practices and problems are of benchmark significance.

Shenzhen is a pioneer city adjacent to Hong Kong on China's southeast coast and is considered to be the first and most successful Special Economic Zone (SEZ) in Mainland China (Figure 1). The Futian CBD is located at the center of the belt-like city, five kilometers west of the Luohu original urban center of the SEZ. It is highly accessible from the rest of the city. Highway, expressway, high speed train, airport express railway, and four additional subways cross the Futian CBD. The adjacent Futian port and Huanggang port gave Futian quick and convenient communication links with Hong Kong, the source of much of the foreign direct investment [27]. Hundreds of modernist skyscrapers are gathered here, with some relatively low-level commercial and service facilities, as well as public spaces, interspersed.

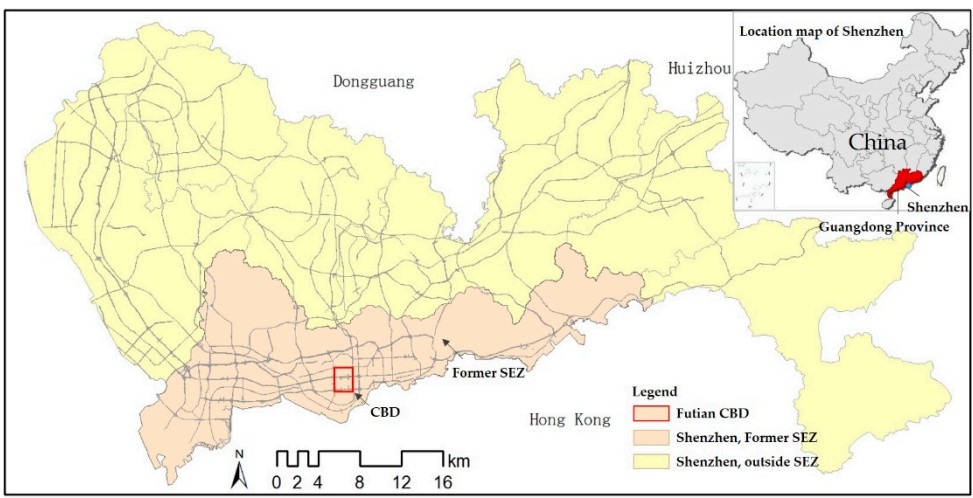

**Figure 1.** A map shows the location of Futian central business district (CBD) in Shenzhen city.

The Futian CBD is built on the lands of a former fishing village and was built up quickly according to a highly formal plan. The 1986 Master Plan of Shenzhen selected the site of the Futian urban center and established its basic concept. A grid network road system was planned and built. The massive construction of business buildings mainly came after the 1998 Asia financial crisis, reaching a peak between 2004 and 2008. By the end of 2014, the total floor area of Futian CBD was 10.9 million square meters, in which 6.6 million was for business (Tables 1 and 2). The city government invested enormous efforts over twenty years to achieve its goal to build a completely new urban center, symbolizing Shenzhen's strategic ambition to be "a modern international city, a core city of the regional economy, a garden city, as well as the city's only financial, business, information, cultural, and administrative center [28]".

**Table 1.** Land use in Futian CBD.

| Items | Total Land Area (m$^2$) | Proportion |
|---|---|---|
| Roads | 1,308,252.0 | 31.8% |
| Greenland & plaza | 737,626.6 | 17.9% |
| Official use | 644,517.5 | 15.7% |
| Residential | 454,988.7 | 11.1% |
| Commercial | 387,585.7 | 9.4% |
| Cultural | 357,679.8 | 8.7% |
| Administrative | 134,307.1 | 3.3% |
| Education & medical | 91,752.4 | 2.2% |
| total | 4,116,710 | 100.0% |

**Table 2.** Floor area of different functions in Futian CBD.

| Item | Total Architecture Floor Area (m$^2$) | Proportion |
|---|---|---|
| Official | 6,588,551 | 60.5% |
| Residential | 2,467,468 | 22.6% |
| Public service | 934,635 | 8.6% |
| Commercial | 818,482 | 7.5% |
| Others | 89,601 | 0.8% |
| Total | 10,898,737 | 100.0% |

Futian CBD, with a total area of about 410 ha, is divided in two by Shennan avenue. The northern part is mainly comprised of official buildings, an administrative center, and cultural facilities, with a total area of 180 ha. The southern part is comprised mainly of business buildings and commercial facilities. A 2000 m long central axis, with a width of 300m to 600m, was located in the middle axis of the CBD. The axis consists of several public buildings and commercial buildings, as well as a platform passing from north to south with connections to the ground floor. The Axis project was first conceptualized in 1985, and construction officially began in 2002. The Axis project was completed in 2010, except for the central part, which was expected to connect the northern part with the southern part of the CBD, as well as the main public buildings.

Non-business functions, including cultural, commercial, administrative, recreational, and educational uses were incorporated into Futian CBD from the beginning. The proportions of land and floor area used for these activities is 23.6% and 16.1%, respectively (Tables 1 and 2). Several shopping malls are located inside the CBD, including a series of commercial complexes, such as the Shopping Park, Central City, Huangting Plaza, Zhuoyue Intown, Dreams-on Store, Central Bookstore, and Link City. However, all of these commercial centers, except the Central Bookstore, are located in the south part of the CBD. In the middle of the CBD is a big park, about 600 x 600 m, divided into two parts by Shennan avenue (Figure 2). To the north of the park, the administrative hall of the city, the Civic Center, which is 470 m long and 160 m width, is situated. Five cultural buildings, namely the Municipal Library, Music Hall, Children's Palace, Urban Planning Exhibition Center, and Contemporary Art Museum, are located to the north of the Civic Center. At the southern end of the north-south axis is the Convention and Exhibition Center, which is used for business conferences and exhibition, also providing catering and recreational services for participants. Other public facility buildings include schools for residents, the children's hospital, up-scale hotels, and so on.

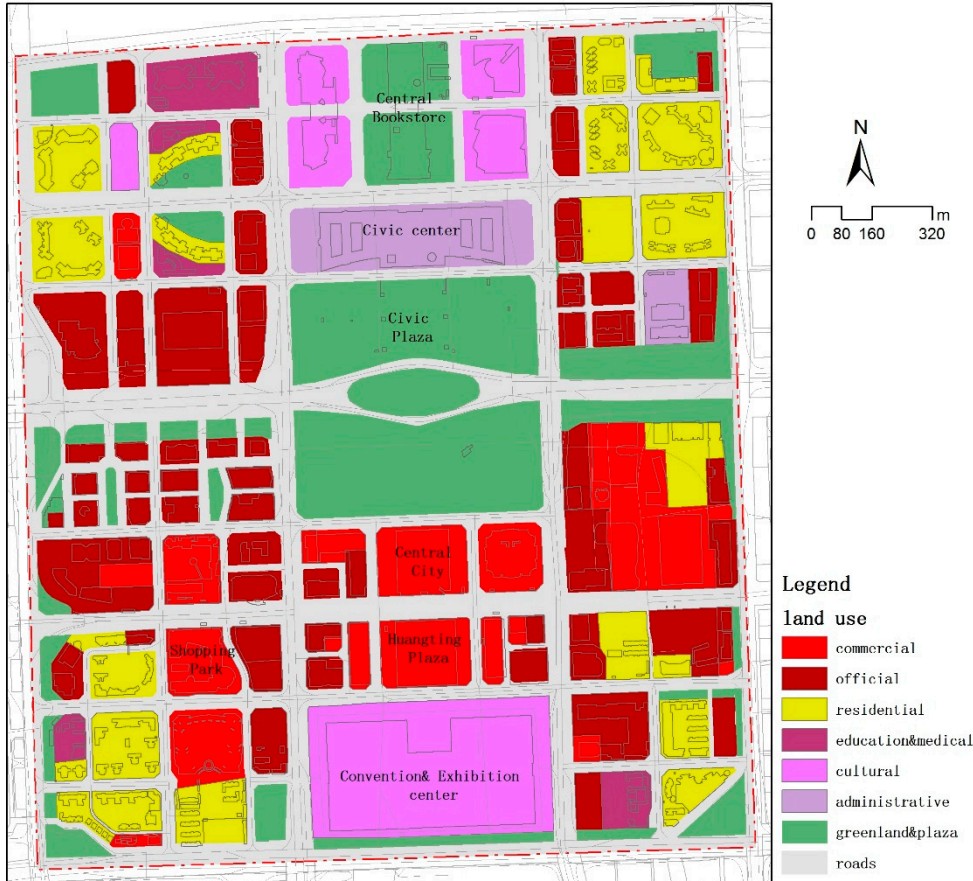

**Figure 2.** Land use map of the Futian CBD.

## 4. Methods

### 4.1. Social Media Material

The service facilities data, which include the attributes of each service facility and user comments, were collected from the following social media network: Dazhong Dianping (www.dianping.com). Dianping is one of the world's largest online and on-demand delivery platforms, reaching up to 10 million daily orders and deliveries (March 2017). It has over 250 million active users and 600 million registered users as of 2016. At the same time, it records over 20 million kinds of shops, including restaurants, recreational facilities, hotels, etc., and covers almost all of China's cities. Users are invited to write comments on the shops and the service after receiving online or offline service.

We develop a web crawler to collect the attributes of all service facilities and their relative user comments. In summary, 13,937 shops with 1,332,819 user comments are collected in Futian district. Among them, 1714 shops with 382,105 user comments are inside the boundary of the Futian CBD. Based on users' comments, the service level of each shop is ranked from one star to five. These data will be used to analyze the composition, distribution, and use of service facilities.

### 4.2. Questionnaire

To understand the composition and experience of CBD visitors, a questionnaire was applied in the Futian CBD. The questionnaire contains two parts: (1) the characteristics of the respondent, including gender, age, occupation, and whether they are living or working in the CBD; (2) the purpose of visiting the CBD, how they came to the CBD, and the reason they choose the CBD for their visit.

The questionnaires are distributed in the major service facility agglomeration spots (including all the main shopping malls) and public spaces in the Futian CBD. At least 30 questionnaires are distributed at each spot. If service facilities at one location are situated at ground level and aboveground or underground level, questionnaires are administered at all levels. The days of survey cover workdays and rest days, including four time periods (Table 3). Five trained university students conducted the field investigation, from 12 August to 21 August 2017. We chose August as it is a typical month with normal weekdays and weekends in a year, it is without long holidays, and the weather is generally fine for off-work activities. After approaching, instructions were provided to survey participants in advance. A small proportion of the potential participants, no more than 5%, refused to continue the survey for time reasons, such as being in a hurry, passing by without time to stay, having to take care of children. This deviation does not affect the representativeness of the sample. Following acceptance, the on-site survey took approximately 5 min to complete. The participants could finish the questionnaire by themselves or with the help of the assistants.

In total, 1204 respondents returned the questionnaire and 1170 (97.2%) were considered as valid. A total of 34 questionnaires were rejected in the analysis because of potential logic errors, including 22 respondents simultaneously working or living in the Futian CBD and seldom visiting the Futian CBD, and 12 answering "live in Futian CBD" for one question and "live in other district" for another question. Among the 1170 valid questionnaires, 53.0% were on workdays and 47.0% were on rest days. The time and space distribution of the questionnaires (Table 3) shows that the investigation covers a wide range of time and various places, and CBD service facility user characteristics (Tables 4 and 5) show the samples are widely representative for the CBD service facility users, including various cultural groups.

**Table 3.** Time and space distribution of valid questionnaires.

| Item | Character | Proportion |
|---|---|---|
| Date | Workday | 52.1% |
| | Rest day | 47.9% |
| Time period | 9:00–11:00 | 7.6% |
| | 11:00–14:00 | 22.0% |
| | 14:00–18:00 | 60.9% |
| | 18:00–20:00 | 9.6% |
| Ground level | Underground | 7.7% |
| | Ground | 78.0% |
| | Above-ground | 14.3% |
| Place | Commercial and catering | 39.8% |
| | Public service venues | 15.3% |
| | Square and green space | 25.9% |
| | Business service | 9.1% |
| | Others | 9.9% |

**Table 4.** Provision and average user comment number of different facilities.

| Items | Description | Counts | Average User Comment Number |
|---|---|---|---|
| Hotel& catering | Hotel, restaurant, coffee, bread, food service | 906 | 401 |
| Commercial& shopping | Official/cultural supplies, clothing shopping, flowers, accessories, digital product, fresh fruit, featured market, jewelry | 347 | 8 |
| Recreational facilities | Attractions, travel agency, culture and art, leisure and entertainment | 125 | 87 |
| Life service | Car services, domestic services, graphic, pharmacy, medical, sports | 168 | 25 |
| Official support | Housing estate service, company, financial, business office, bank, government agency | 132 | 3 |
| Traffic service | Traffic connection, station, traffic service, parking lot | 37 | 10 |

**Table 5.** CBD facility users' characteristics.

| Item | Character | Proportion |
|---|---|---|
| Gender | Male | 51.0% |
| Age | <18 | 8.7% |
| | 18–24 | 30.8% |
| | 25–30 | 34.7% |
| | 31–50 | 22.2% |
| | 51–60 | 2.1% |
| | >60 | 1.5% |
| Travel mode | Subway | 67.6% |
| | Bus | 21.4% |
| | Car | 9.1% |
| | Taxi | 7.6% |
| | Bicycle | 11.3% |
| | Walk | 17.2% |
| Travel time | <30 min | 46.4% |
| | 30–60 min | 36.9% |
| | 1–2 h | 10.8% |
| | >2 h | 5.9% |
| Purpose of visiting CBD | Business | 14.3% |
| | Shopping | 22.8% |
| | Recreation | 36.0% |
| | Cultural/sports | 16.8% |
| | Socializing | 22.4% |
| | Eating | 35.5% |
| | Accommodation | 3.3% |
| | Others | 14.9% |
| Workplace | Futian CBD | 30.0% |
| | Futian district, Non-CBD | 15.6% |
| | Other district | 54.4% |
| Resident place | Futian CBD | 23.8% |
| | Futian district, Non-CBD | 16.0% |
| | Other district | 60.2% |
| Visiting frequency | Every day | 24.4% |
| | 1–2 times a week | 24.4% |
| | 3–6 times a week | 17.0% |
| | Monthly | 18.1% |
| | Seldom | 16.2% |
| Occupation | Government official | 6.2% |
| | Financial& insurance | 21.5% |
| | Professional technology &consulting | 23.9% |
| | Retail &food service | 8.5% |
| | Others | 39.9% |

*4.3. Regression*

A binary logistic regression analysis was used to investigate the association between potential predicting factors of whether a service facility user is a visitor or business traveler. The dependent variable is whether the user is a visitor or not, while the independent variables include whether it was a workday or not, whether it was daytime or not, travel mode, ground level, and facility location. Considering the spatial planning characteristics of the Futian CBD, the service facility location attributes are divided into five categories, as follows: 1 large shopping center; 2 large cultural facilities; 3 large green spaces or squares; 4 residential units; 5 office buildings. The results are presented in terms of

significance ($p < 0.05$), odds ratios (OR), with 95% confidence intervals (CI). The goodness-of-fit of the models is assessed by the χ-square test. SPSS Version 22 (IBM) was used for all statistical analyses.

## 5. Results

### 5.1. Distribution and Vitality of Service Facility

Although various service facilities are indispensable functions for the CBD, the supply of service facilities is not uniform and fully market-oriented. The function of urban land in China is strictly regulated. Commercial facilities and cultural facilities are concentrated on commercially zoned and culturally zoned land. Business space usually does not provide service facilities, and there are few service facilities in residential plots.

The distribution of facilities is generally uneven in the Futian CBD (Figure 3). Although the plan designated a large number of cultural and commercial plots for development, the existing facilities are concentrated within the southern part of Futian. The huge number of service facilities and use intensity reflects the high service concentration of the southern part of the Futian CBD. In the northern part, however, the facility density is much lower and unevenly distributed, with only a few scattered facilities and without a shopping center or mall (Figure 2). On plots developed for business use, there are almost no service facilities. The Central Bookstore, where dozens of shops and restaurants are collected, provides a limited number and types of commercial services; however, it does so far away from the official buildings. In a word, the provision of service facilities is agglomerated and uneven, especially in the northern part (Figure 3). This outcome reflects a lack of attention to the development of such facilities by the government, even if the urban plan allowed them in the first place.

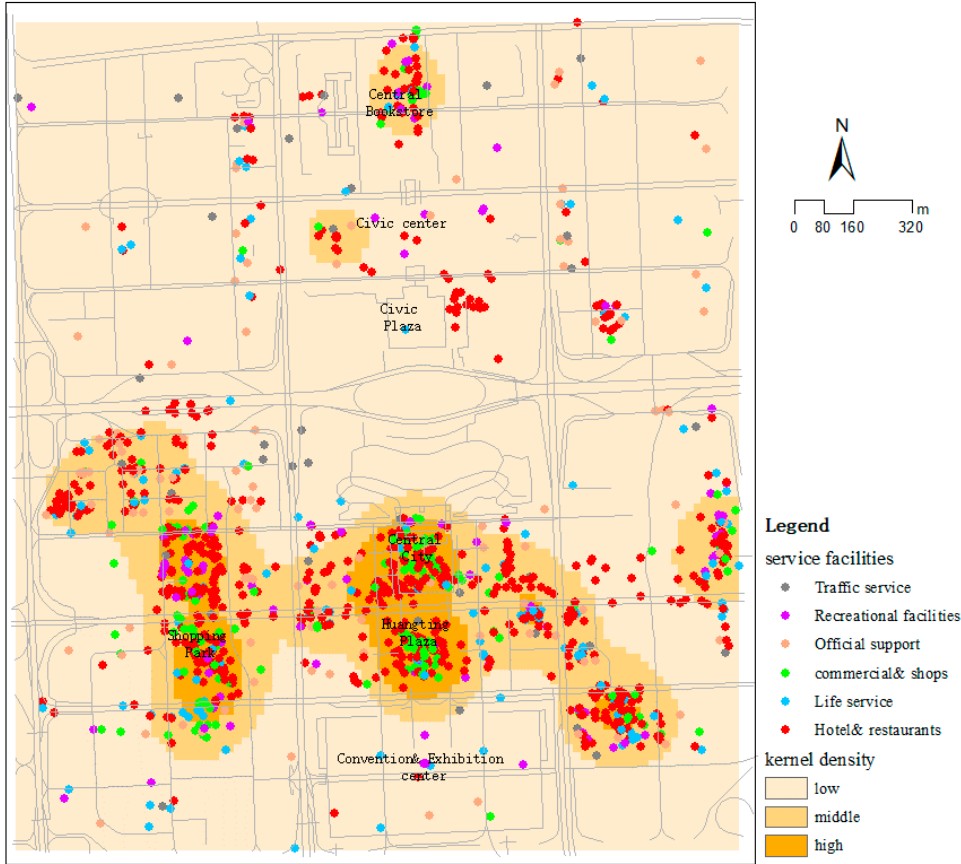

**Figure 3.** Distribution map of service facilities in the Futian CBD.

Here we see two different distribution models of service facility in the Futian CBD: concentrated commercial business facilities and dispersed support facilities. To investigate how these service facilities are used and the intensity of use, we employ the number of user comments from social media to compare vitality between different types of facilities.

Figure 4 shows the average comment number of all facilities, and Table 4 shows the average comment numbers of each facility type. In the CBD, hotel and catering facilities have the highest number of comments, while official supportive facilities have the least. In terms of spatial distribution, the number of comments on facilities in the south is obviously more than that on the north. The facilities with the highest vitality concentrate in the commercial facility concentrated areas (Figure 1). In other areas, and for other service facilities, the use vitality is fairly low. It is obvious that the concentrated facilities contributed most to the vitality of the service facilities.

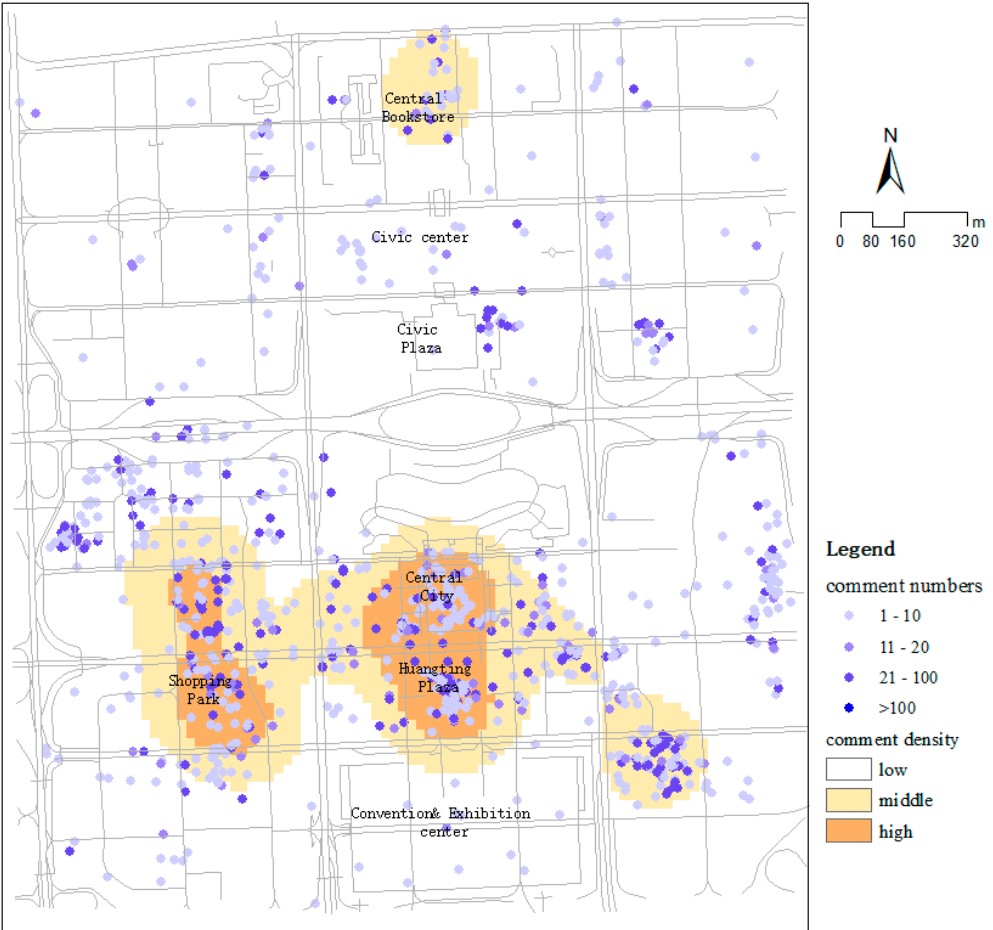

**Figure 4.** User comment number at service facilities in the Futian CBD.

## 5.2. User Characteristics

The CBD is a highly complex urban center with numerous functions, focusing on intensive business use. The total employment within the Futian CBD is more than 400,000 [28], which is ten times the employment density of the SEZ. A large number of service facilities are planned to serve this employed population. At the same time, in order to build a more popular and modern CBD, The Futian CBD integrates functions such as high culture venues, governmental services, and commercial services in a pitch at image-building and the appearance of diversity. These functions each occupy a large independent plot, in contrast with neighboring Hong Kong, where many service facilities are located within office buildings, the podium of the skyscraper, and in underground space.

These independent land-based services not only serve CBD workers but also attract visitors from other parts of the city. According to user characteristic statistics from our questionnaire (Table 5), CBD facility users are significantly younger than the population of the city [29]. In other words, mainly young people are using CBD facilities.

In terms of the arrival travel model (Table 5), the majority of CBD facility users choose public modes. Specifically, 67.6% of CBD facility users travel by subway and 21.4% by bus. From the beginning of the Futian CBD's development, public transit systems were placed in the most important position [28]. About 60% of the respondents reside in districts other than Futian, and 16.7% of the users take more than one hour to travel to the CBD. Only 24.4% make daily visits to the CBD. These characteristics indicate that the Futian CBD is not only a local center or a business center, but an important civic center that attract visitors from far afield.

The reasons for visiting the CBD are rich and varied, including business, shopping, recreation, entertainment (including tourism, sports, and socializing), eating, accommodation, and other. Only 14.3% of the users claim a business purpose; in other words, the vast majority of CBD facility users are engaged in non-business activities. Recreation (36.0%), eating (35.5%), shopping (22.8%), and socializing (22.4%) are among the top four activities by popularity. About half (48.0%) of the respondents claim more than one purpose for visiting the CBD. Thus, the CBD serves not only as a place for business but also as an interesting destination for multipurpose non-business activities.

Of all the CBD facility users, only 30.0% claim to work in the CBD, compared with 54.4% claiming to work in districts other than the Futian district. This astonishing result reveals that the majority of the CBD facility visitors do not work in the CBD. Considering the widely recognized business role of CBDs, we see a new civic role of Futian's CBD, which attracts many people with a non-business purpose to use the service facilities. Besides the central location of the Futian CBD, the relation between the distribution characteristics of facilities and user behavior needs further investigation.

*5.3. Use Pattern: Business Travelers and Non-Business Travelers*

Initially intended for the convenience of business travelers working in the CBD, the service facilities are shown to attract a lot of non-business travelers based on the results of our questionnaire. According to the purpose for visiting the CBD and their workplace, the facility users are divided into two categories: business travelers and non-business travelers (Figures 5 and 6). Business travelers, comprising those who work in the CBD or come for a business purpose, account for 34.5% of the respondents. Non-business travelers refer to other respondents.

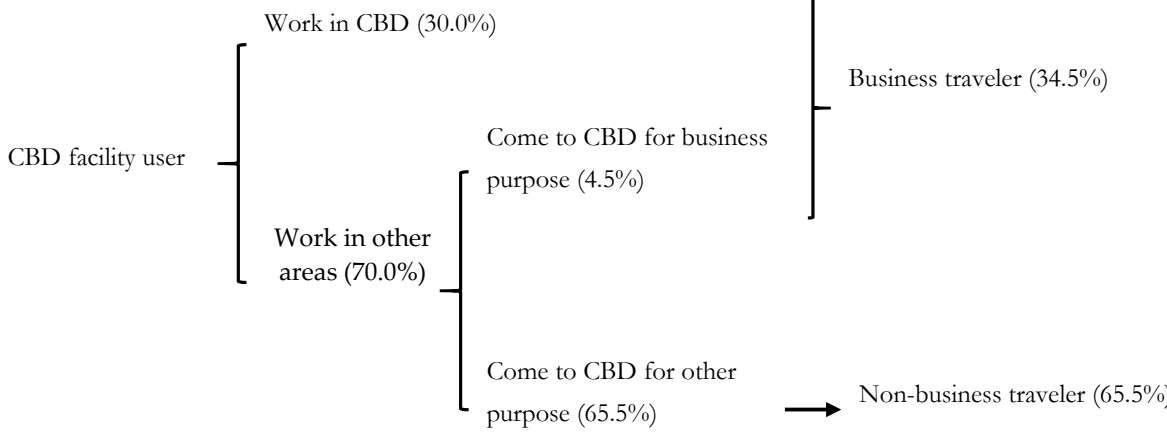

**Figure 5.** Classification of business travelers and non-business travelers.

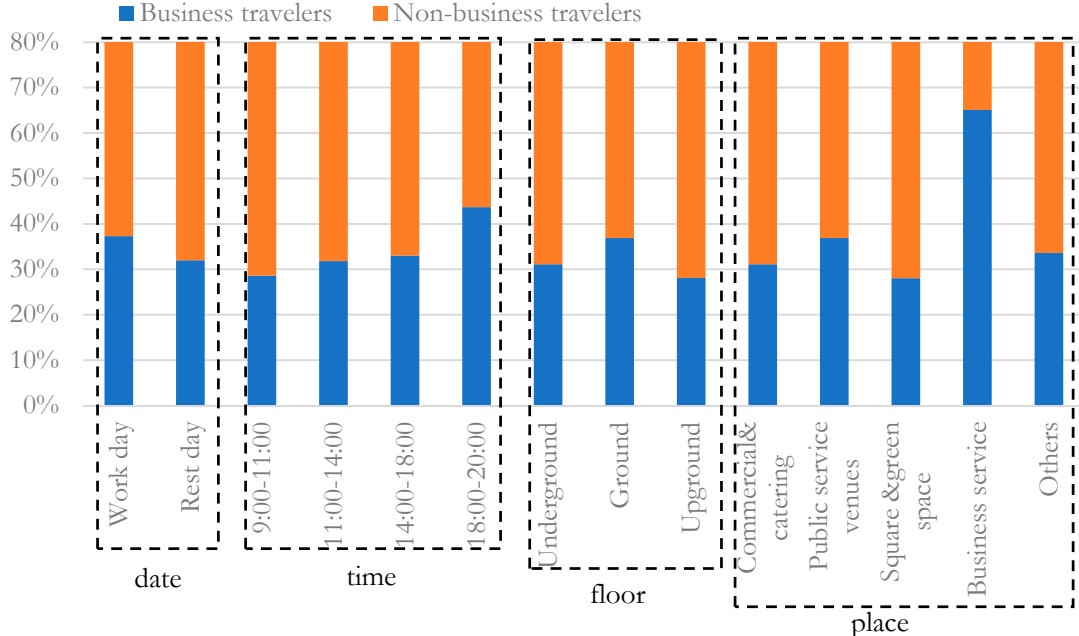

**Figure 6.** Proportion of business travelers and non-business travelers.

There are far more business travelers than non-business travelers in the Futian CBD. Why do the service facilities attract so many visitors? First of all, the Futian CBD has unique location conditions. It is located in the geometric center of the original SEZ and has excellent traffic accessibility. Secondly, the Futian CBD has introduced large-scale city-level cultural facilities, municipal centers, large shopping centers, and large municipal squares. The civic plaza has become a public gathering place where many large public events take place. For example, the recent lighting show on weekend nights and holidays attracts hundreds of thousands of visitors. The Library and Book Plaza has also become a popular destination for citizens on weekends. The shopping centers in the south area of the CBD all attract plenty of consumers at both weekday nights and weekends.

Spatially, CBD facility users mainly congregate in the southern part of CBD, where most of the service facilities are provided. Taking Shennan avenue and the main roads as separating lines to divide the CBD into six sections (Figure 7), there are very few service facilities in the northern part of the CBD. Specifically, in section A and section C, the density of service facilities is the lowest (Figure 3), resulting in the lowest user density (Figure 7). The proportion of non-business travelers in sections A and C is much lower than it is in other sections. The land use map (Figure 2) shows that in sections A and C there are mainly residential and business plots, with a few commercial plots. This unbalanced spatial distribution of land use leads to the unbalanced distribution of facility users and activity vitality.

What drives service facility users to be CBD visitors? In order to explore the impact of service facility characteristics on CBD users, a binary logistic regression model was prepared. The regression results (Table 6) show that the service facilities of large shopping centers, cultural facilities, and large green squares are more likely to be used by visitors. The odds ratios are 1.288, 2.063, and 1.493, respectively. The service facilities in office buildings and residential communities are more likely to be used by business travelers. It is the large shopping center, cultural facilities, and the large green spaces introduced in the central area of Futian that attract a large number of non-businesspeople to visit the CBD.

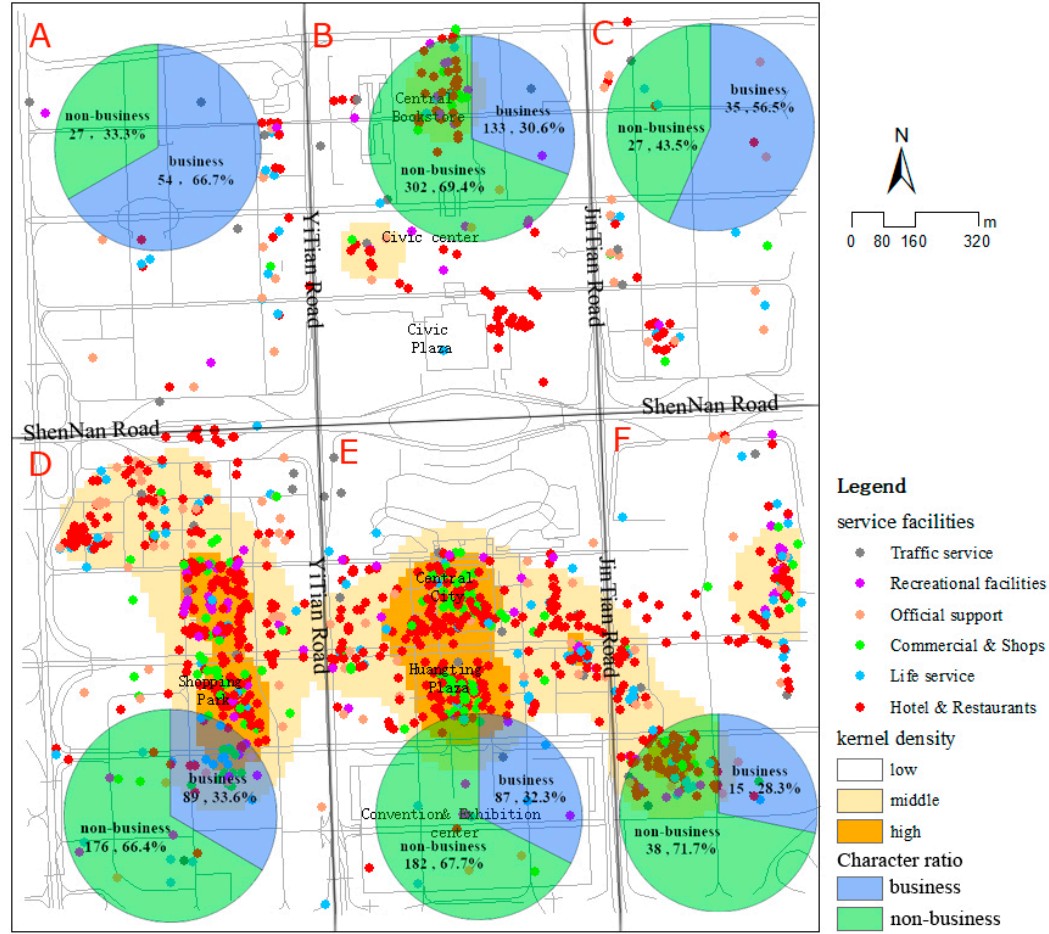

**Figure 7.** Spatial distribution of CBD facility users' density.

**Table 6.** Odds ratio of non-business traveler over business travelers for CBD facilities use.

| Item | Varies | OR | 95% Lower | 95% Upper |
|---|---|---|---|---|
| **Day** | Rest day (work day = 1) | 1.354 * | 1.063 | 1.725 |
| **Time** | Daytime (night time = 1) | 1.368 * | 1.042 | 1.797 |
| **Travel mode** | Subway | 1.722 ** | 1.337 | 2.219 |
| | Bus | 1.396 * | 1.029 | 1.894 |
| | Car | 0.953 | 0.629 | 1.445 |
| | Taxi | 0.887 | 0.566 | 1.388 |
| | Bicycle | 0.635 * | 0.437 | 0.922 |
| | Walk | 0.535 ** | 0.393 | 0.729 |
| **Ground level** | Ground level(yes=1) | 0.731 * | 0.568 | 0.939 |
| | Underground level(yes = 1) | 1.119 | 0.707 | 1.771 |
| | Aboveground level(yes = 1) | 1.397 | 0.973 | 2.006 |
| **Facility location** | Shopping center(yes = 1) | 1.288 * | 1.005 | 1.653 |
| | Cultural center(yes = 1) | 2.063 ** | 1.333 | 3.193 |
| | Square and green space(yes = 1) | 1.493 ** | 1.121 | 1.988 |
| | Residential community(yes = 1) | 0.601 ** | 0.410 | 0.881 |
| | Official building(yes = 1) | 0.288 ** | 0.203 | 0.410 |

(* $p < 0.05$, ** $p < 0.01$).

Different users are not evenly distributed in time or space. In a day, the proportion of business travelers among CBD facility users grows gradually from morning to afternoon and reaches its peak in the evening (Table 6). Non-business travelers are more likely to visit the CBD on rest days (OR = 1.354) and the daytime (OR = 1.368). This is a clear piece of evidence that the CBD's service facilities have greatly boosted the vitality on rest days. Business travelers are more likely to use ground floor facilities (OR = 1.368). This result indicates that the underground and aboveground transit systems needs improvement, especially convenient walking connections. In terms of travel mode, business travelers are more likely to choose a bicycle (OR = 1.575) or walk (OR = 1.868), while non-business travelers are more likely to use public transportation, such as a bus (1.396) or the subway (OR = 1.722). The mass transport system contributes greatly to attracting non-business travelers.

## 6. Discussion

In the context of globalization, as the economic structure of cities gradually becomes more diversified, the functions of urban centers also tend to diversify. After the urban development in North America experienced the wave of urban sprawl and suburbanization, CBD development faced many problems. Recently, its citizens are leading an exodus from suburbia back into the urban core, resulting in the revitalization of the downtown areas [30,31]. A lot of downtown revitalization projects that incorporated urban retailing were trying to attract shoppers back to the CBD [31]. Many Chinese megacities began to build a CBD from the mid-1990s, undergoing a rapid physical growth after 2000 in a handful of cities [15].

A variety of service facilities are important to business travelers of the CBD. On the one hand, the CBD is the region with the highest employment density in the city. It needs high-end service facilities to serve the needs of white-collar workers during work, including high-quality space for face-to-face communication, diverse public spaces, high-end restaurants, entertainment, and culture facilities. On the other hand, high-end shopping, culture, entertainment, catering, and other service facilities are needed to serve the needs of CBD workers after work [32]. In Hong Kong's CBD, a large number of service facilities have been deployed around the two subway stations in Admiralty and Central, which satisfy both the needs of the visitors and the workers [33].

However, in the service facilities of the Futian CBD, non-commercial travelers have become the dominant user group. Despite the large number of service facilities planned, our survey shows that two-thirds of the users are non-business travelers. Numerous factors have resulted in service facilities attracting a large number of visitors. High accessibility contributed greatly to the attractiveness of the CBD. Large shopping malls, cultural facilities, and large green squares are incorporated into the CBD but they are mostly used by visitors. In terms of space, the commercial facilities are concentrated in the southern part of the CBD, and the cultural facilities are mainly located on the northern part; however, these facilities are without a convenient connection to the business plots. The center of the CBD is a large formal square, designed to serve as a symbolic setting for the urban center, not as a place for business gathering. During the daytime, there are "no citizen in the civic plaza"; only at nighttime or on major events, such as anniversaries, does the square would attract many visitors. Moreover, the addition of this large-scale axis square has interrupted the east–west communication scheme, thereby making crossing communication inconvenient [27].

At the surrounding area of the Futian CBD, much high-end residential use is planned. In the office centers of most western cities, there is little or no residential and ancillary services. But because of traffic congestion and high commuting costs, high-end homes around the office center have begun to emerge. Those successful commercial centers are surrounded by high-end residential areas, such as the New York and Chicago office centers, which are housed by high-income staff [34]. In China's new CBDs, Beijing, Guangzhou, and Shenzhen have planned some high-end residential areas, equipped with supporting facilities. In the case of Futian, these new planning concepts have increased the variety and vitality of the CBD, especially on weekends and weekday nights. Our survey shows that the main

users of these high-end residential areas and ancillary services are CBD workers, which means that these high-end residential areas mainly serve the business activities.

The CBD is a multi-functional mixed area dominated by business and commercial buildings, whereas the non-central area of a city is mainly focused on living and recreational functions. As a vibrant part of the city, the CBD will change dramatically over time [35]. Economic globalization means that even the megacity no longer serves only the interests of CBD workers; it also serves visitors' interests. The Futian CBD is changing from accommodating business activities to major service areas. From the comparison of facility provision, it was found that the commercial and cultural facilities in Futian have changed the role of the CBD to some extent. Futian's CBD has shifted from a business center to a civic center. By linking multiple functional areas with a CBD's business functions, Chinese cities have successfully expanded the reach of the CBD. Almost everyone, and not only businesspeople, can get some public services in the CBD. The CBD has become an upgraded version of the city center for the general public.

Considering the decline and revitalization process experienced by CBDs in western countries, the urban plan of the Futian CBD has tried to avoid the problem of insufficient nighttime vitality and insufficient services through the introduction of commercial and cultural facilities. However, the introduction of too many concentrated commercial and cultural facilities in Futian has also caused some disruption to the CBD, resulting in a large number of CBD visitors, including many visitors to civic cultural facilities, and recently, crowding at the lighting shows. The influx of large-scale populations has also brought about serious traffic congestion and the problem of insufficient facilities during certain time periods. To this end, the Futian CBD has specifically established a management agency and added many temporary commercial and service facilities to deal with the management and security issues of a large number of tourists. At the same time, in view of the lack of connection between commercial cultural space and office space, the management agency is also studying to build a closer relationship between different functional land through the two-story corridor system and underground transportation system.

Regarding the future development of CBD, finding a balance between serving workers and visitors is a question that needs further study in future CBD plans. In essence, the investment in the CBD needs to seek a return, and the return rate of the office building is stable and low. The injection of business and service functions has brought new vitality to the sustainable development of CBD. The nature of work is also changing. Consumption-oriented activities and social communications are playing more and more important roles. Nowadays, the office trend is moving towards the virtual office, part-time office, and shared office mode, and a lot of work can be done outside traditional work units. The office of a company becomes a place for meetings and exchanges. The accessibility of an office space, the provision of certain cultural service facilities and modern business service facilities have become very important. We suggest that in the business plots in the CBD, more service facilities are needed to meet the needs of the businesspeople. At the same time, convenient pedestrian passages are suggested to connect the existing commercial district and the business district and promote the service ability of the commercial facilities to the office workers, both at work and off work. It is difficult for urban planning to consider these complex issues and uncertainties. In order to cope with the uncertainty of future urban development and the changes in transportation methods, the planning of the CBD should also adopt a "blank" approach to leave room for future development.

## 7. Conclusions

The CBD has become the economic powerhouse of contemporary cities. China's economic transition from world factory to a knowledge-based economy underpinned the development of hundreds of CBDs over the course of less than two decades. The plans promoted land use diversity and the incorporation of service facilities in the support of business function, but a rather different service environment emerged.

Taking the Futian CBD of Shenzhen as the prototypical case, we examined the distribution, vitality, uses, and users of these facilities, which were largely built up by the private sector and without governmental support. A questionnaire for users and data derived from social media reveal that the vast majority of visitors to these service facilities do not work in the CBD and travel via the reformed mass transport system to this location. We found that the service facilities in China's new CBDs have played an urban center role—not solely serving business travelers but attracting a lot of non-business travelers as well. The large worker population spends little time in the facilities of the CBD, departing for other locations at the end of the workday. The high-quality public spaces and street environment, as well as the numerous service facilities, many of which are at a low economic order, attract people from all over the vast city, highlighting a new role for the CBD as a civic center. The central location of the Futian CBD, as well as the introduction of various service facilities, is an important reason for the large number of visitors to the Futian CBD.

In contrast with the globalized business sought after by government and business leaders for the CBD, a new populist nexus is emerging and without significant support. We suggest that in the business plots of a CBD, strong connections are needed between the existing commercial district and the business district, and more service facilities should be added in business plots to promote the service ability of CBD.

**Author Contributions:** Investigation, M.Z.; Methodology, Y.C. and J.Z.; Project administration, J.Z.; Writing—original draft, Y.C.; Writing—review & editing, J.Z. All authors have read and agreed to the published version of the manuscript.

**Funding:** This research was funded by the Open Fund of Key Laboratory of Urban Land Resources Monitoring and Simulation, Ministry of Land and Resources.

**Conflicts of Interest:** The authors declare no conflict of interest.

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
