# Peer review of "Searching for the Center: A New Civic Role for the Central Business District in China"

_sustainability, doi:10.3390/su12030866_

Round 1

Reviewer 1 Report

Dear Authors,
The subject is very interesting. The research method does not raise any objections. The results of the research and discussion are on a high scientific level. My only comment concerns the literature review. You can add more current items. In my opinion the article has the potential to be published.

Author Response

we thank the anonymous reviewer very much for reading the manuscript so carefully, and offering this important comments. We have add more current literature and made relative revisions in the manuscript.

Reviewer 2 Report

There are some things that are unsettling about this paper. Nowhere is there a clear sense of what the paper is all about. The title is misleading. Questions were thrown here and there without any specif reference to the question(s) the paper seeks to address (see examples in lines 71, 117, 239, etc.). The paper rambles on about CBDs in what is the introduction but should be part of the literature review. Terms such as command function, representational, facilities, etc., were used without normative or operational definitions. Sweeping comparisons were made between CBDs in the West and in China without strong supporting evidence or data. It was not until line 120 that the reader gets a sense of the area of study, which is really a unique case in China from which generalizations cannot be sensibly drawn. Lines 402 - 412 capture what this paper is all about; a descriptive study of patrons of Futian CBD and the activities (functions or land uses) at this CBD. The paper does not explain or identify, as the title and some of the questions suggested, WHY the intended purpose of these planned Chinese CBDs changed from business to what it is today, i.e., a combination of civics, leisure, etc. The discussion and conclusion were borderline syllogistic and do not gel with the title, aim and questions of the paper. The authors should simply tell a story from the data they analyzed and show how their story provides insights on CBDs in Shenzhen, and perhaps China. 

Author Response

we thank the anonymous reviewer very much for reading the manuscript so carefully, and offering so many important and constructive comments. These criticisms are sharp but meaningful. We have tried to make various changes to make the research theme more prominent and the logic more clear. We reorganized the theoretic and literature review part, especially the research background and our study question.

Please allow us to take this opportunity to send our greatest appreciation. We have benefited tremendously from the constructive comments and suggestions, which are all extremely helpful to bring the paper up to the standard and for our future researches.

Reviewer 3 Report

This is a valid topic to be used, i.e. what happens in the major Chinese cities' CBDs, and how th planning of the uses and the actual uses meet or do not meet each other.

The manuscript could be improved with paying attention to a few issues:

1) validity of the survey data - is August particular season for the CBD use? Who probably did not answer to the survey? How about the "alternative" groups useing CBD (such as yougters, different gangs, artistic groups, minorities, etc.)? Does this make a difference thinking about the planning of CBD use and the actual (measured) use?

2) is the case something particular in China or a typical case? why so? how?

3) how about some references to former socialist cities elsewhere and the CBD development?

4) perhaps some attention is needed to think about the future of CBDs - can it be based on consumption-oriented activities? how about urban sustainability? how about post-private car development?

5) does the potential conflict between planning (for business) and actual use (leisure) create any probmes for the city authorities, and more broadly to the planning rationale of CBDs in China?

6) newer references to the contemporary use of CBDs (and their revitalization) is needed, also aoutside the USA

Author Response

To reviewer: we thank the anonymous reviewer very much for reading the manuscript so carefully, and offering so many important and constructive comments.

We have benefited tremendously from your constructive comments and suggestions, which are all extremely helpful to bring the paper up to the standard and for our future researches. Please allow us to take this opportunity to send our greatest appreciation to you.

1) validity of the survey data - is August particular season for the CBD use? Who probably did not answer to the survey? How about the "alternative" groups useing CBD (such as yougters, different gangs, artistic groups, minorities, etc.)? Does this make a difference thinking about the planning of CBD use and the actual (measured) use?

Response: we thank the reviewer for these important comments. We add detailed information of the survey and discussed its potential influence in section 3.2.

2) is the case something particular in China or a typical case? why so? how?

Response: we thank the reviewer for asking this very important question. We described why we chose the case and its typical meaning in section 2.

3) how about some references to former socialist cities elsewhere and the CBD development?

Response: we thank the reviewer for this suggestion. We have add some references of former socialist cities and made some comparisons in introduction section and literature section.

4) perhaps some attention is needed to think about the future of CBDs - can it be based on consumption-oriented activities? how about urban sustainability? how about post-private car development?

Response: we thank the reviewer for this very important suggestion. We add some discussion on the future of CBD in the last paragraph of section 5.

5) does the potential conflict between planning (for business) and actual use (leisure) create any probmes for the city authorities, and more broadly to the planning rationale of CBDs in China?

Response: we thank the reviewer for this excellent question. We add discussions in last paragraphs of section 5.

6) newer references to the contemporary use of CBDs (and their revitalization) is needed, also aoutside the USA

Response: we thank the reviewer for this suggestion. We add one paragraph in section 1 to discuss the contemporary use of CBDs and their revitalization.

Round 2

Reviewer 2 Report

There is a 'message' in this paper that is misplaced, miscommunicated or lost somehow.

What question(s) did this research pose and sought to answer? How did the information collected answer the question(s), hence, addressed the research aim(s)? Where in the conclusion is the closing of the loop, which clearly states that the research question(s) was/were answered by the data collected?